# Densification of SiCp/Al–Fe–V–Si Composites by the Wedge Rolling Method

**DOI:** 10.3390/ma16124290

**Published:** 2023-06-09

**Authors:** Yiqiang He, Changbao Huan, Qianhang Su, Lijie Zuo, Wen Feng, Yunfei Ding, Feng Shang, Yi Luo, Lichao Feng, Yan Wang, Hang Gu

**Affiliations:** 1College of Mechanical Engineering, Jiangsu Ocean University, Lianyungang 222005, China; 2021210201@jou.edu.cn (C.H.); 2020210202@jou.edu.cn (Q.S.); 2022210503@jou.edu.cn (L.Z.); 2021210213@jou.edu.cn (W.F.); 2020221323@jou.edu.cn (Y.D.); 2020221301@jou.edu.cn (F.S.); jou_zn@163.com (Y.L.); 2020221319@jou.edu.cn (L.F.); 2020000073@jou.edu.cn (Y.W.); gh141131@163.com (H.G.); 2Jiangsu Marine Resources Development Research Institute, Lianyungang 222005, China

**Keywords:** spray deposition, wedge pressing, densification, al-matrix composite, microstructure

## Abstract

The densification of a SiCp/Al–Fe–V–Si billet was achieved by reducing the pores and oxide film between the particles by rolling. The wedge pressing method was used to improve the formability of the composite after jet deposition. The key parameters, mechanisms, and laws of wedge compaction were studied. The results showed that the pass rate was reduced by 10 to 15 percent when using steel molds during the wedge pressing process if the distance between the two ends of the billet was about 10 mm, which was beneficial to improve the compactness and formability of the billet. The density and stress of the surface of the material were higher than those of the interior, where the distribution of density and stress tended to be uniform as the overall volume of the material shrank. During the wedge extrusion process, the material in the preforming area was thinned along the thickness direction, while the material in the main deformation area was lengthened along the length direction. Under plane strain conditions, the wedge formation of spray-deposited composites follows the plastic deformation mechanism of porous metals. The true relative density of the sheet was higher than the calculated value during the initial stamping phase, but was lower than the calculated value when the true strain exceeded 0.55. This was due to the accumulation and fragmentation of SiC particles, which made the pores difficult to remove.

## 1. Introduction

As early as 1986, Skinner et al. [1] of Allied Signal Company in the United States developed heat-resistant Al–Fe–V–Si aluminum alloys by using plate flow casting (PFC) technology. It was used for temperatures up to 400 °C, and was considered an alternative to titanium alloys for high temperature applications. The Al-8.5Fe-1.3 V-1.7Si (type 8009) alloy was also considered a promising candidate for high temperature applications. At present, Al–Si alloy is widely used in electronic packaging, aerospace, and other fields, as per Li et al. [2] and Jia et al. [3]. In particular, alloys reinforced with dispersion precipitation, grain reinforced alloys, or whisker reinforced alloys were found to be more attractive [4,5,6,7].

Multilayer jet deposition technology combined with ceramic particle co-deposition has become a powerful tool for the development of high-performance composite materials. The resulting materials have fine grains and exhibit excellent strength and stiffness, strong wear resistance, high temperature stability, and high density, as per Yan et al. [8], Tian et al. [9] and Zhang et al. [10] prepared an Al–12Si alloy by jet deposition and hot rolling. By changing the content of Mn and Mg in the alloy, the comprehensive properties of the material were improved, while the fine precipitated phase led to a significant increase in tensile strength. During the deposition of aluminum alloy and composite materials by spray, there were some pores and oxide films with different shapes in the powder matrix. For example, Xiang et al. [11] reported three main types of pores with porosities ranging from 1% to 10% for various spray-deposited aluminum alloys. Better-performing materials can also be obtained by changing the spray parameters in the process of material preparation. Chen [12] changed the spraying speed to prepare the surface layer of cast copper, and found that under high current density and scanning speed and low spraying speed, the deposited grains were finer, while the structure was denser. Pores formed in the deposited billet could be divided into three types: interstitial type, one type resulting from atomizing gas, and one type resulting from the solidifying contraction. Nucleation and growth of the pores can be stretched.
(1)$$P_{g}-P_{s}\geq P_{i}+\frac{2\sigma }{r_{p}}$$
where *P_g_*, *P_s_*, and *P_i_* are the evolutionary pressure, contraction pressure, and ambient pressure, respectively; 2*σ* is the surface tension of the hole in the material; and *r_p_* is the radius of the hole. According to Formula (1), the porosity of the billet is related to the dissolved gas in the melt. The inherent porosity in spray-deposited materials leads to two kinds of problems. First, mechanical properties (especially at high temperatures) might degrade due to the presence of pores. Second, the deformation ability of prefabricated parts is obviously reduced, especially for composite materials. As a result, Jin et al. [13], Nouri et al. [14], and Zhang et al. [15] proposed that secondary working such as stirring or extrusion would be necessary to reach full density. Laminate rolling technology and frame rolling technology were used to densify the deposited SiCp/Al composites, which improved the microstructure and mechanical properties of the composites from the experiments of Wei et al. [16] and Zhang et al. [17]. The SiCp/Al–Fe–V–Si composite sheet was prepared by extrusion and hot pressing, and the SiC particle distribution and SiC–Al interface were studied by Jiao et al. [18] and He [19,20,21]. However, due to the poor deformability of jet deposition material, it was difficult to roll directly. In the process of extrusion, the difference of flow velocity between matrix particles and SiC particles easily led to the stratification and aggregation of SiC particles. The ceramic rolling technique, frame-pack rolling technique, and hot pressing were restricted from densifying preforms of large size because of the high resistance to deformation at elevated temperatures, limited tonnage of equipment, and high cost. Through experimental calculation and analysis, Shu et al. [22] studied the material in the process of hot rolling, and found that the cooling speed caused by the processing sequence also has a great influence on the performance. Li et al. [23] studied a TC6 alloy prepared by wedge rolling. The properties of the alloy were improved by the change of equiaxed grain volume, grain refinement, and the precipitated phase. These factors significantly improved the tensile strength of the material. Lian et al. [24] carried out finite element simulation on a billet rolled by a wedge. The variation of grain size and recrystallization in the rolling process of the billet was analyzed, and the variation rule of average grain size after a time change was obtained.

In this paper, the porosity of the material was reduced by adjusting the technological parameters of the wedge extrusion. Wedge rolling had a unique advantage to improve the densification and deformation ability of large size composites. Using low tonnage equipment, the porosity of large size spray deposited SiCp/Al composites without SiC particle aggregation could be reduced at relatively low cost, and the deformation ability could be improved. The densification mechanism and law were studied, and the deformation simulation was carried out.

## 2. Experimental Procedures

### 2.1. Preparation of Materials

The nominal composition of the Al–Fe–V–Si alloy was Al–8.5Fe–1.3V–1.7Si. SiC particles with a volume fraction of 15% and an average particle size of about 3 μm were selected as the reinforcement phase. The experimental process parameters of the spray deposition are shown in Table 1.

### 2.2. Wedge Pressing and Rolling

The wedge press mainly consisted of three parts: the wedge punch (on a hydraulic press with a capacity of 6300 kN), the cast steel frame, and a heating device. First, the heating element and external frame were placed on the operating platform. During extrusion, the angle between the wedge forming plane and the compression plane was 5~6°, which provided a good space for predeformation during the deformation process. During the extrusion process, the workpiece and the cast steel frame remained motionless. The punch was lifted after each squeeze, moved forward to the next position, and pressed again. Through repeated extrusion, the material was thinned and densified. The workpiece was annealed for 20 min during each extrusion to proceed to the next round of compaction. Thus, jet-deposited slabs with near-full density can be obtained by multiple passes and local small deformations of the extrusion pile-up. The device and extrusion principle of the wedge extrusion process are shown in Figure 1.

The workpiece was extruded and densified at 450~480 °C, while the workpiece temperature was monitored by a thermocouple. The billet and mold were heated to 480 °C and held for 60 min before wedge pressing. The compaction and thickness reduction rate of the sheet preblank was 10~30%. The most important parameters in the machining process include the thickness reduction rate *ε*, the wedge pressure temperature, and the overall deformation of the material. In order to avoid extrusion cracking of the billet, the thickness reduction rate of each extrusion pass must be less than 10% at the initial stage of wedge extrusion. Then, as the workpiece density increases, the thinning rate can be appropriately increased. Before extrusion, the material was preheated at a given temperature (480 °C) for 1 h, and the thinning rate of the material was set to 10%. Al–8.5Fe–1.3V–1.7Si/SiCp composites were extrusion-insulated for 10 min at 480 °C each time, and the thickness of the material was 0.6~0.8 mm after multiple cycles.

### 2.3. Material Characterization and Testing

Tensile specimens with the tensile axis parallel to the machine direction were machined from rolled and pressed composite sheets. The mechanical properties of the composites were tested by a UTM5305 electronic universal testing machine in a tensile test at a tensile strain rate of 5 × 10^−4^.

The microstructure of the composites was observed by optical microscopy (OM, XMP-50, Shenzhen City, Guangdong Province, China)), SEM (FEI Verios 460 model, Hillsboro, OR, USA), and TEM (Talos L120C, Hillsboro, OR, USA). The phases of the composites in different states were determined by a D8A X-ray diffractometer (Cu target, 60.0 kV, 80.0 mA) from Bruker AXS (Billerica, MA, USA).

### 2.4. Finite Element Simulation

The size of the preform for FEM modeling was 150 mm (length) × 110 mm (width) × 20 mm (thickness), and the initial relative density of the porous preform was about 87%. The modeling temperature, the linear expansion of the preform, the coefficient, the elasticity modulus, the thermal conductivity, and the specific heat capacity were set as 480 °C, 2.08 × 10^−5^/K, 107 GPa, 95 W/m·K, and 903 J/kg·K, respectively. The pressing velocity of the punch (regarded as a rigid material), was set as 1 mm/s, and the thickness reduction rate was set as 5%. The heat exchange with the ambient condition was defined in Heat Exchange with Environment, and every surface of the preform was regarded as a heat exchange surface. First, the preform was fixed with the mold, and then the contact between the preform and the mold was established. When the heat transfer coefficient was set to 11 W/(m^2^·K), the friction coefficient was set to 0.3 with a plastic shear friction model.

## 3. Results and Analysis

### 3.1. Wedge Pressing

Figure 2 shows a simulation of blank deformation with or without a gap between the blank and the mould. There was no space between the precast and the mould, which caused the billet bulge problem. Too little extrusion led to uneven deformation of the billet surface and inside, and too much extrusion led to cracks on the billet surface. Wedge pressing elongated the precast in its length direction, and the end of the precast needed to be left with a space between the steel mould to prevent arching and cracking of the precast.

As shown in Figure 3, this can be explained by the flow behavior of the prefab during the wedge extrusion. From the preformed zone to the strain zone, which was from point P1 to point P4, the flow rate gradually decreased. This change can be attributed to the fact that the relative densities of the precast regions at points P1 and P2 were lower than those at points P3 and P4 in the strain region. In the red dashed line, the pores in the precast region collapsed and bridged, while the densification behavior of the precast region was higher than that of the strained region at strong compressive stress. The material in the precast zone flowed down the thickness direction to reduce the thickness of the slab. The material in the strain zone flowed along the length direction and the slab length increased. At the beginning of the extrusion, the slab thickness decreased due to the reduction of pores, and the slab length hardly increased (when the overall volume decreased by less than 20%). When the overall volume of the material was reduced by more than 20%, the slab length increased significantly.

At the beginning of the extrusion, the thickness of the slab decreased and the length of the slab almost did not increase due to the destruction of pores (when the total reduction was less than 20%). When the material flowed along the length and the overall reduction was greater than 20%, the slab length increased significantly.

### 3.2. Density Distribution

Figure 4 shows the morphologies of different parts of the material at a 20% thickness reduction rate. Figure 4a,c show that some collapse pores and elongated holes on the surface of the slab. Wedge extrusion formed 20 μm long strip holes distributed in the middle region (Figure 4b).

Figure 5a represents the density differentiation from the upper layer to the middle layer and then to the bottom layer. If the material was squeezed by the top and bottom molds, the SiC and precipitated particles were easy to aggregate, resulting in high density in the upper and bottom layers. In the wedge pressing process, the density distribution of the material tended to be uniform with the decrease of the overall volume and the increase of the extrusion amount. As can be seen from Figure 5b, the microstructure of the composite was also improved after wedge-shaped extrusion. The SiC particles inside the material were evenly distributed and the pores were reduced, while the performance of the material was improved to a certain extent.

Figure 6 shows the DEM simulation results of the prefab density distribution when the total quantity was reduced by 20%. It can be found that during the extrusion process, the relative density of the upper layer was 0.891, which was higher than that of the middle layer of the composite. Moreover, the relative density decreased from the surface to the center. This can be attributed to the gradual reduction of the pressure stress from the surface to the interior.

### 3.3. Densification Mechanism and Regularity

The elongation of the billet in the width direction was limited by the steel mold. The results show that the slab width remained unchanged, but the billet length increased and the thickness changed with the increase of the total extrusion pressure. Thus, when a wedge pressure was applied, the material deformed under compressive stress.

*σ_n_* (*n* = 1, 2, 3) and *ε_n_* (*n* = 1, 2, 3) are the primary stress and principal strain in three directions, respectively. From the studies of Yi et al. [25] and Kuhn et al. [26], it can be deduced from the mass conservation of porous metal matrix material during plastic deformation:(2)$$d\varepsilon_{1}+d\varepsilon_{2}+d\rho /\rho =0$$

Due to the pressure and friction of the die and wedge, the billet was in a state of three-way compressive stress. According to the yield function of porous material, it can be deduced as per Kuhn et al. [26]:(3)$$f={\left[\frac{{\left(\sigma_{1}-\sigma_{2}\right)}^{2}+{\left(\sigma_{2}-\sigma_{3}\right)}^{2}+{\left(\sigma_{1}-\sigma_{3}\right)}^{2}}{2}+(1-2\nu_{\rho })(\sigma_{1}\sigma_{2}+\sigma_{2}\sigma_{3}+\sigma_{1}\sigma_{3})\right]}^{\frac{1}{2}}$$
where *f* and *vρ* are the yield function and the Poisson’s ratio of porous material.

The elastic potential was introduced by Mises into the plasticity theory. A plastic potential function *K*(*σij*) similar to the elastic potential function was assumed in the plastic flow, and the flow direction was consistent with the gradient direction of the plastic potential function *K*. Subsequently, as per Hu et al. [27], the potential theory of metal plastic forming was obtained:(4)$$d\varepsilon_{\mathrm{ij}}{}^{p}=d\lambda \frac{\partial K}{\partial \sigma_{i}{}_{j}}$$
where *K*(*σij*) is a function of primary stress *σ*_1_, *σ*_2_, and *σ*_3_. The *dλ* is a nonnegative proportionality coefficient. The strain increment stress equation was obtained by the differential of the yield function.

The density relation of a porous metal material under planar strain conditions can be formulated as follows from He et al. [28]:(5)$$\varepsilon_{n}=\frac{1}{4}\ln(1-\rho^{2})-\ln\rho +C$$

Formula (5) is the densification behavior of the loose metal material under the plane strain condition, where the constant *C* is determined by the initial relative density. By substituting *C* into Equation (5), the calculated relative density of the SiCp/Al–Fe–V–SiC material during spraying and pressing can be obtained. The comparison between the calculated relative densities of the composite and the experimental results during the extrusion are shown in Figure 7.

As shown in Figure 8, the relative density of the prefabricated parts increased with the increase of the true stress. The measured relative density of the material was higher than that calculated by the finite element simulation. This means that the true density and intensities of the material were higher than the theoretical values during the initial stages of wedge squeezing. When the true strain *ε* was greater than 0.16, the densification rate decreased. When ε was greater than 0.55, the experimental relative density was smaller than the calculated relative density.

When the actual strain increased, the experimental density decreased and the theoretical density increased. This can be explained as follows. On the one hand, the density increased dramatically due to the collapse and closure of the pores. On the other hand, the theoretical value of the material was based on the stress and strain in the downward direction, while the force and strain in the width direction of the billet during extrusion was not considered. In fact, the strain in the width direction existed to a certain extent during the wedge pressing.

In Formula (5), the fact that the addition of SiC particles made the theoretical density different from the experimental density was ignored. SiC particles can reduce the densification in the last stage of wedge pressing. It came down to three points. Firstly, it was difficult to eliminate the pores between the aggregated particles in the extrusion process. Secondly, the anti-sticky property between the SiC particles and Al matrix led to the formation of pores. Thirdly, the extrusion of silicon carbide particles resulted in cracks and holes. These factors caused the theoretical relative density to be higher than the calculated value.

### 3.4. Microstructure

Figure 9 shows the microstructure of spray deposited SiCp/Al–8.5Fe–1.3V–1.7Si and wedge pressed SiCp/Al–8.5Fe–1.3V–1.7Si and the SiC–Al interface after extrusion.

Figure 9a–c show the evolution of the microstructure during wedge extrusion. In the process of the atomizing deposition, the cooling rate of 100 μm diameter droplets deposition reached up to 2.55 × 104 K/s, and fine particles were obtained. As can be seen from Figure 9a, SiC and Al_12_(Fe,V)_3_Si reinforced particles with diametric ranges from 60 nm to 150 nm were distributed in the Al matrix. As shown in Figure 9b, SiC reinforced particles and Al_12_(Fe,V)_3_Si particles maintained good particle morphology during extrusion. There was no obvious deformation or crack in the particles, indicating that the SiC reinforced particles maintain good stability. A clean, smooth, and uniform Al–SiC interface with a width of about 3 nm can be observed in Figure 9c. Nanoscale Al–8.5Fe–1.3V–1.7Si precipitated particles could alleviate the interface non-wetting characteristics between the ceramic particles and metal matrix to a certain extent. The binding effect of the Al matrix and SiC particles was improved. The mechanical properties of the composites were also improved.

XRD in Figure 10 shows that the material was mainly composed of SiC, α-Al, and α-Al_12_(Fe,V)_3_Si. There was no significant θ-A1_13_Fe_4_ in the XRD pattern, indicating that the α-Al_12_(Fe,V)_3_Si remained stable and did not transition to θ-Al_13_Fe_4_. A large number of stable precipitates (i.e., Al_12_(Fe,V)_3_Si precipitates) with a high fraction (usually 20–30%) were formed during jet deposition because SiC reinforced particles can maintain a stable strengthening effect at high temperature for a long time, while the heat resistance and strength of composites were improved.

Due to the slow cooling rate, coarser tissue was formed under pouring conditions. The microstructure of Al–8.5Fe–1.3V–1.7Si was related to the cooling rate. Al_13_Fe_4_ and Al_3_FeSi formed the equilibrium phase when the cooling rate was lower than 103 K/s, while α-Al_12_(Fe,V)_3_Si formed the equilibrium phase when the cooling rate was higher than 103 K/s. Therefore, the cooling rate of up to 104 K/s during jet deposition was conducive to the formation of α-Al_12_(Fe,V)_3_Si. If the composites were heated at 599 °C for a long time, the silicide Al_13.18_(Fe,V)_1.84_Si particles with a lattice parameter of 1.2578 nm could still strengthen the alloy and would not change to Al_13_Fe_4_ (Yaneva et al. [29]).

### 3.5. Fracture Surface

The larger extrusion stress and deformation were conducive to the metallurgical bonding between the deposited particles, thus improving the comprehensive properties of the materials. Park et al. [30] found that, in an RS Al–Fe–V–Si alloy (0.3–0.7 μm), the sizes of the pits formed on the surface of the material were distributed in the range of 0.3–2 μm, and the smaller pits were consistent with the grain size. Previous studies have shown that Al_12_(Fe,V)_3_Si grains after heat treatment were mostly located at grain boundaries. Arhami et al. [31] suggested that the fine pits may be caused by Al_12_(Fe,V)_3_Si particles in the grain. The Al_12_(Fe,V)_3_Si was formed through cavity nucleation, growth, and coalescence, while the matrix was then shear deformed to form the tough nests. Figure 11 shows the tensile fracture morphology of the sample after rolling at different temperatures. In Figure 11a, a large number of equiaxial dimples appeared at the fracture of the tensile specimen, but no separation of holes and interfaces was observed. This indicated that the pores and microcracks were subsequently bridged or eliminated by wedge pressing and rolling.

The tensile fracture of the material was observed at room temperature (26 °C). During loading, the stress accumulation near the SiC particles was much higher than that around the matrix. The SiC particles were fractured by force and the fracture surface was smooth. As shown in Figure 11a, the SiC particles remained closely bonded to the matrix after stretching, while the enlarged image shows that there was no crack at the bonding interface of the material. Figure 11b shows the tensile test results of the material at 100 °C. By observing the fracture surface, it was found that the SiC particles were still in close contact with the matrix. The enlarged image proves that there was a strong bond between the particles inside the material. As the tensile temperature increased, the number of broken particles decreased and the number of separated particles increased. If the tensile temperature was increased to 200 °C, as shown in Figure 11c, cracks appeared and expanded between the SiC particles and matrix, as shown in the enlarged image, which confirmed the increasing trend of particle detachment within the material. Figure 11d shows the tensile fracture of the material at 300 °C. With the increase of temperature, the binding force between the SiC particles and matrix became weaker, while the pits and cracks of the material increased, and the SiC particles were broken less. The enlarged image shows the pores formed by the shedding of SiC particles. The stress on the particles in the material exceeded the bonding strength between the SiC particles and Al matrix.

The degradation of the properties of the SiCp/Al composites was mainly due to the fracture of reinforced SiC particles, the cracking and spalling of the interface between the SiC and Al matrix, and the stress fracture of the Al matrix. As can be seen from Figure 11, with the increase of tensile temperature, heat provided the driving force for particle movement. Driven by energy and stress, the movement of the SiC matrix and Al matrix led to the fracture and crack propagation of interfacial bonding. When the temperature was lower than 200 °C, the binding between SiCp/Al was relatively tight, and the SiC particles fractured under the action of tensile stress.

The fracture of SiC particles was mainly caused by the thermal stress inside the material, the stress concentration around the particles, and the natural brittleness of SiC particles. The crack formation rate was related to the length–diameter ratio of SiC particles, the interfacial shear strength, and the distribution of SiC particles in the metal matrix. In the process of low temperature tensile deformation, the larger SiC particles broke first, and the smaller SiC particles broke later. The SiC and matrix still maintained strong interfacial bonding. At temperatures above 200 °C, the debonding of the composite rolling interface was the main mechanism of composite fracture. As the temperature increased, the crack in the SiCp/Al interface expanded, and the shedding of SiC particles was more serious than the crack.

The position of SiC particles after shedding formed dimples. SiC particles prevented the flow and elongation of the Al matrix during extrusion, and the stress around the SiC was concentrated to produce tearing edges after shedding. With the increase of temperature, the uneven distribution of SiC intensified the interface stripping and particle shedding. The uniform distribution of SiC particles and stress can be improved by controlling the extrusion pass, extrusion amount, and wedge extrusion mode. The interfacial bonding strength inside the material was improved, and the properties of the material were improved.

## 4. Conclusions

The large SiCp/Al–Fe–V–Si slab formed by jet deposition was densified by the wedge rolling method. The following conclusions can be drawn:(1)Wedge compaction can effectively reduce the porosity of preformed parts and improve the metallurgical bonding between deposited particles in preformed parts.(2)In the wedge extrusion process, the volume thinning rate of SiCP/Al–Fe–V–Si deposited by jet deposition should be reduced by 5~15% compared with that of the raw billet. The material was effectively densely packed inside. The material in the deformation zone flowed along the length direction and the length of the deformation zone increased. In addition, as the material flowed along the thickness direction, the thickness of the precast area decreased.(3)The density was not uniform from the surface to the center, which was consistent with the finite element simulations. In the wedge-shaped extrusion process, the uneven density distribution was gradually improved with the increase of the total extrusion amount.(4)After extrusion at 480 °C, Al_12_(Fe,V)_3_Si particles maintained well. The Al–SiC interface was clean, smooth, and uniform, and the width of the interface was about 3 nm. This nanoscale interface was expected to improve the wettability of SiC in the Al matrix.(5)The fracture mode of rolled composites was related to the interfacial strength, which was related to the tensile temperature. When the tensile temperature was lower than 200 °C, the cracking of SiC particles was the main fracture mode, and when the tensile temperature was higher than 300 °C, the matrix–particle interface shedding was the main fracture mode.

## Figures and Tables

**Figure 1 materials-16-04290-f001:**
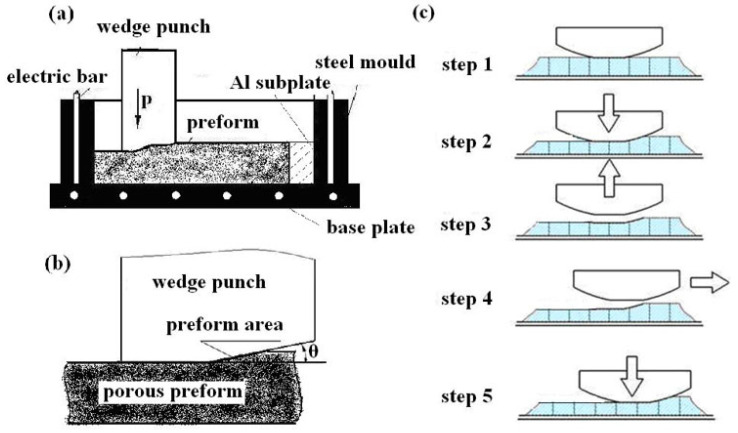
Wedge extrusion process diagram: (**a**) device, (**b**) pressing theory, and (**c**) pressing process.

**Figure 2 materials-16-04290-f002:**
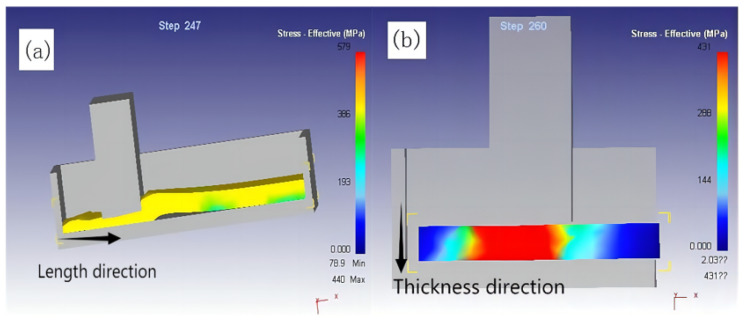
Deformation of the preform (**a**) with and (**b**) without space between the preform and the mold.

**Figure 3 materials-16-04290-f003:**
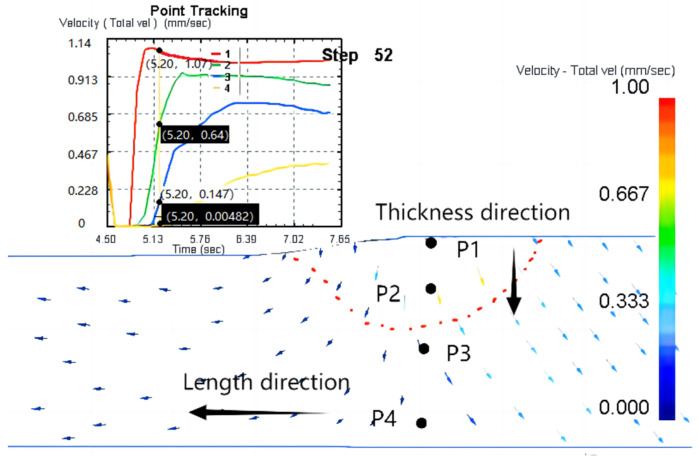
Flow characteristics of wedge forming billet.

**Figure 4 materials-16-04290-f004:**
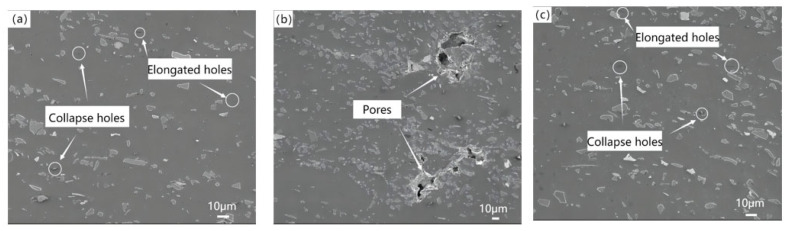
Microstructure of the preform with thickness reduction rate of 20% in (**a**) upper surface layer, (**b**) interior region, and (**c**) the lower surface layer.

**Figure 5 materials-16-04290-f005:**
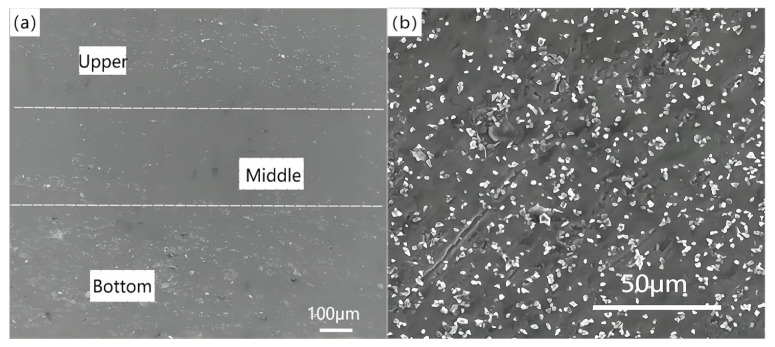
(**a**) Density of inner area of material and (**b**) microstructure of wedge-rolled sheet.

**Figure 6 materials-16-04290-f006:**
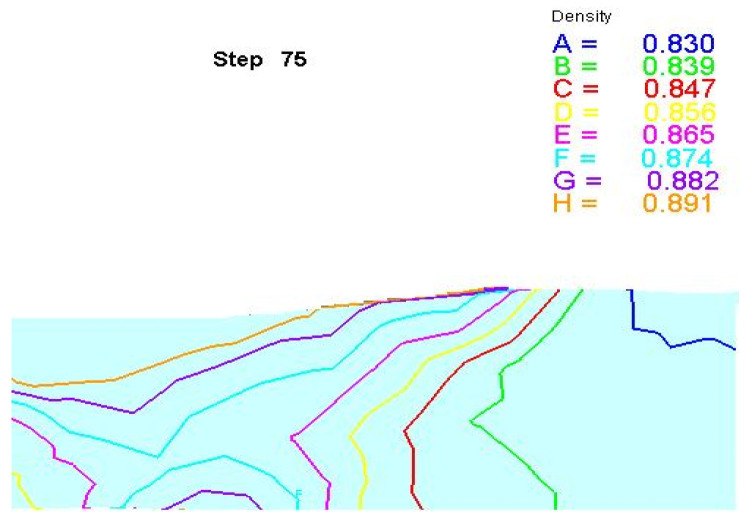
The total reduction of the relative density of the slabs during wedge pressing was 20%.

**Figure 7 materials-16-04290-f007:**
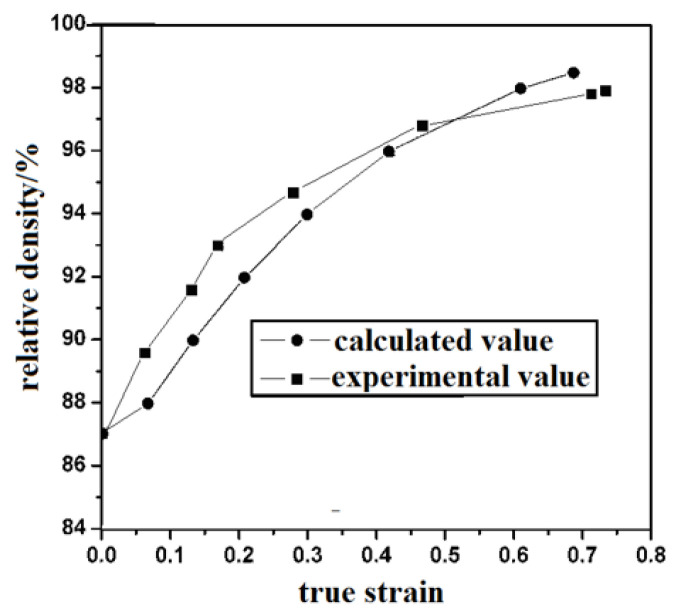
Calculated and experimental values of the relative density of recombination during extrusion process.

**Figure 8 materials-16-04290-f008:**
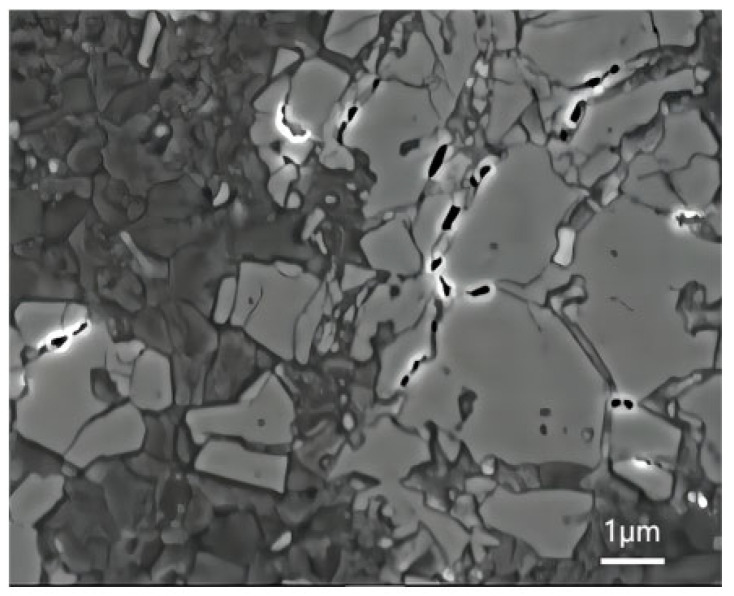
Aggregation and fragmentation of SiC particles (true strain ε = 0.6).

**Figure 9 materials-16-04290-f009:**
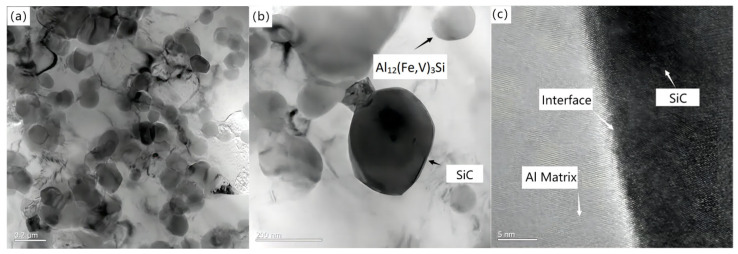
Microstructure of Al-8.5Fe-1.3V-1.7Si and its composite: (**a**) as-spray deposited, (**b**) as-wedge pressed, and (**c**) SiC–Al interface.

**Figure 10 materials-16-04290-f010:**
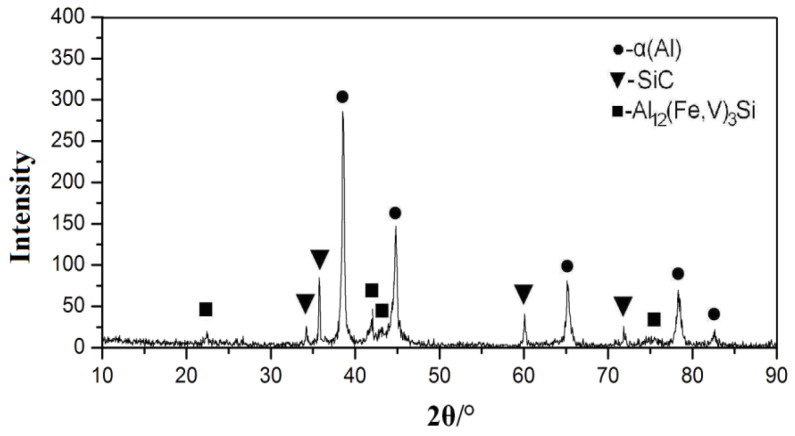
XRD of SiCp/Al–8.5Fe–1.3V–1.7Si composite as spray deposited.

**Figure 11 materials-16-04290-f011:**
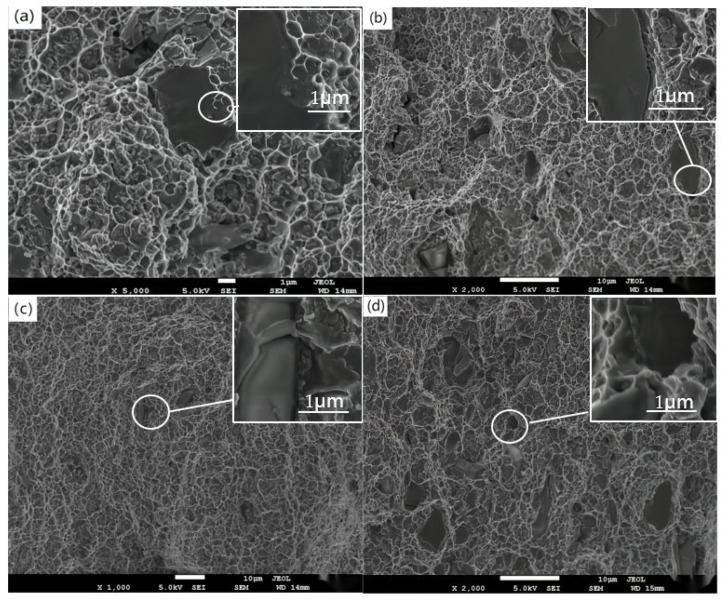
Tensile fracture surface of rolled SiCP/Al–8.5Fe–1.3V–1.7Si at different tensile temperatures: (**a**) ambient temperature, 26 °C; (**b**) 100 °C; (**c**) 200 °C; and (**d**) 300 °C.

**Table 1 materials-16-04290-t001:** Process parameters of the spray deposition experiments.

Experimental Parameters	Values
Vaporization temperature/K	1223~1373
Diameter of the melt stream/mm	3.2~3.6
Spray height/mm	200~350
Rotation speed of the substrate/r·min^−1^	100~350
Scan velocity of the nozzle/s	10~30
Pressure of the atomization gas/MPa	0.7~0.9

## Data Availability

No new data were created or analyzed in this study. Data sharing is not applicable to this article.

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
