# Peer review of "Densification of SiCp/Al–Fe–V–Si Composites by the Wedge Rolling Method"

_materials, 2023, doi:10.3390/ma16124290_

Round 1

Reviewer 1 Report

The manuscript is incomprehensible. The authors have many interesting results to present but due to the poorly written manuscript it is very difficult for the author to read it. Is is thus suggested to try to write it again and explain step by step the experimental procedure. Afterwards, present the results in a clear and consistent manner.

The use of English in this manuscript is very poor. It really needs a lot of work and improvement.

Author Response

1. English has been rewritten and revised.
2. The introduction and abstract have been revised.
3. The references have been revised.
4. The experimental procedure and conclusion have been revised.

Reviewer 2 Report

This work reports a study densification of metal matrix composites by wedge rolling method.

Introduction: This part of paper need attention. Introduction must provide a comprehensive critical review of recent developments in a specific area or theme that is within the scope of the journal (The of Materials). I personally feel that this part of paper is not describe enough from a reader's perspective.  Besides, literature review is better to present relevant researches in the latest years.

The Authors should include some data on new processing and modification, such as laser surface remelting or laser-assisted machining (doi.org/10.3390/ma15238375, doi.org/10.1051/matecconf/201821902005, DOI: 10.1007/s00170-017-1304-z, DOI: 10.1051/matecconf/201713602006).

Methodology and discussion are curried out correctly in my opinion.

I think the paper can be accepted for publication after consideration of the Reviewers suggestions

English language is correct

Author Response

(The authors gave the same response as above.)

Reviewer 3 Report

Thanks to the authors for submitting the paper to the journal Materials!

The paper focuses on the densification of SiCp/Al-Fe-V-Si composites by the wedge rolling method. The subject of the study as a whole is of scientific and practical importance for the development of production technology and modification treatment of heat-resistant aluminium alloys of the Al-Fe-V-Si class, in which pores and oxide films tend to form during densification. The authors draw attention to the possibility of using available low-tonnage equipment to reduce the porosity of large-size SiCp/Al-Fe-V-Si composites deposited by sputtering without particle aggregation. The mechanisms and regularities of densification studied are presented in the paper. The authors have modelled the deformation process.

The work deserves to be presented to the scientific community and published in Materials. There are the following minor remarks and comments on the paper related to the formatting. :

 – the authors sent the manuscript in a sloppy form, which partly makes it difficult to read the material. For example, most of the drawings in the paper are in need of considerable improvement in terms of image quality. In particular, the text captions in many of the figures are indistinguishable. In Figure 4, it is almost impossible to figure out the text with the size of the scale bars. The authors should look at the layout of Table 1 and re-organise it so that it is clear which value goes to which row. There is a typo in the name of the system, "SiCP/al-fe-v", on line 141. As well as many other errors in organising text and graphics.

– authors are encouraged to increase the specificity of the abstract using the traditional approach to its development: objective, design/methodology/approach, results/implications, significance/relevance.

Author Response

1. English has been rewritten and revised.
2. The introduction and abstract have been revised.
3. The references have been revised.
4. The experimental procedure and conclusion have been revised.
5. Figures and text have been revised.。

Reviewer 4 Report

Abstract language not clear, example paragraph in line number 12 to 14, “pass-through reduction” and “while the steel die”, “The density and stress of the surface layer”, and in line number 17 “area flowed downward along the thickness direction”.

The Introduction started directly with the literature review; it would be better if the author introduces this article work what is it about? The problem statement or which deposition technology they used??

Line 51 “And 2σ/ rp are the radius and surface tension of spherical pores, respectively.” This paragraph needs rearrangement of the variables.

 Experimental procedure section is missing proving some information about the deposition technology in use, although the table 1 provided the process parameters.

Not clear line 104 “According to the results, the slab and die were heated to 480℃,and the workpiece was insulated for 60 minutes before wedge pressing.”

Unusual manufacturing process terminologies or process parameters used in this section, such as “the thickness reduction rate, ε ”, “thinning rate”, and describing this hot press process as “extrusion”. Read in line 102 to 111.

Results and analysis: line 150-151 ”As mentioned above, 150 there was no space between the preform and the mold, which led to the bulging problem”. Why was this problem not addressed in the experiment design although it is expected to happen?

Line 155-157 “This can be explained by the flow behavior of preforms during wedge extrusion. Fig.3 shows that the flow rate gradually decreased from the pre-cast region to the strain region.”  Please identify these two regions (pre-case and strain regions) on the supplied figure 3.

Line 160, “This change was due to the fact that the relative density and intensity of the pre-160 formed region was lower than that of the strained region.” Not clear the term “intensity” used in this paragraph meant what? is it the stress intensity? Or other property??

Line 169 using the term “destruction of pores” can be better described as “annihilation” or “elimination”.

Please identify the “thickness direction” and “length direction” in any suitable image provided.

Section “3.2. Density distribution” Figure 4 scalebars not clear. Line# 178

Line 180 the use of term “density” in describing porosity reduction is confusing.

Provide references for equation (2) to (8).

Line 297-298, “Park et al. (1994) used to think in RS Al-Fe-V-Si alloy (0.3-0.7μm), the dimple size was distributed in the range of 0.3-2μm, and the smaller dimple size was consistent with the grain size.” Not clear paragraph in saying “used to think in” and the “dimple” term.

General comments:

-          The article used excellent characterizations and modelling simulations tools.

-        The language of the article is not clear, and the narrative is not concise, this could be due to the complexity of the researched project, but as a suggestion better explaining and diagrams can be added explaining the deposition technology used and the wedge extrusion gadget with details of the terms used in text.

-         Discuss the importance of this project and its applications please in the introduction and conclusions.

The English should definitely be improved. There were sections of the paper which were very hard to understand.

Author Response

1. English has been rewritten and revised.
2. The abstract has been revised.
3. The introduction has been revised.
4. The experimental procedure has been revised.
5. The expansion problem in lines 150-151 of the results and analysis refers to the non-use of wedge extrusion;
Figure 3 has been labeled; line 160 has been revised; line 169 has been corrected; Figure 4 has been drawn; equations have been revised with the addition of references; lines 297-298 have been corrected.

Round 2

Reviewer 1 Report

The manuscript is now improved and ready for publication

Author Response

The manuscript was revised based on the comments, please see the attachment.

Reviewer 4 Report

The authors have addressed most of my comments. I have no further feedback.

There are still a lot of grammatical errors that needs to be corrected before publication.

Author Response

(The authors gave the same response as above.)
